# On the Numerical Modeling of Flax/PLA Bumper Beams

**DOI:** 10.3390/ma15165480

**Published:** 2022-08-09

**Authors:** Liu Jiao-Wang, José A. Loya, Carlos Santiuste

**Affiliations:** Department of Continuum Mechanics and Structural Analysis, Universidad Carlos III de Madrid, Avenida de la Universidad 30, 28911 Leganés, Spain

**Keywords:** green composite, bumper beam, numerical modeling, energy absorption, natural fibers

## Abstract

Significant progress has been made in green composites developing fully biodegradable composites made of microbially degradable polymers reinforced with natural fibers. However, an improvement in the development of numerical models to predict the damage of green composites is necessary to extend their use in industrial applications of structural responsibility. This paper is focused on developing a numerical model that can predict the failure modes of four types of bumper beams made of flax/PLA green composites with different cross sections. The predictions regarding energy absorption, contact force history, and extension of delamination were compared with experimental results to validate the FEM model, and both results revealed a good agreement. Finally, the FEM model was used to analyze the failure modes of the bumper beams as a function of the impact energy and cross-section roundness. The impact energy threshold defined as the maximum absorbed-energy capability of the beam match with the impact energy that produces delaminations extended through all the cross sections. Experimental and numerical results revealed that the threshold energy, where the maximum energy-absorption capability is reached, for Type A is over 60 J; for Type B and C is around 60 J; and for Type D is at 50 J. Since delamination is concentrated at the cross-section corners, the threshold energy decreases with the cross-section roundness because the higher the roundness ratio, the wider the delamination extension.

## 1. Introduction

Plastics and composites are ideal materials in the automotive industry for interior, exterior, and structural applications [1]. Composites present advantages in terms of specific mechanical properties and they are competitive in manufacturing cost; however, composites are difficult to recycle. The creation of the fully biodegradable composite material, also called “green composites”, can counter this disadvantage of composites with respect to metals. Green composites are essentially composites made of natural fibers and microbially degradable polymers. Their mechanical properties are quite different from those of carbon- or glass-fiber-reinforced polymer composite materials. Elastic nonlinearity and strain-rate dependency features make this green material more difficult to analyze [2]. Therefore, more comprehensive research on automotive components made of fully biodegradable composite becomes essential.

Among the vehicle protective components, bumper beams remarkably affect the structural energy-absorption capacity to protect the occupant and pedestrian under low-speed impacts [3]. Bumper beams can be one of the most suitable car components to be manufactured using natural fiber-reinforced polymers (NFRP). There are many types of green composites and NFRPs that can be applied to the automotive industry. Some of the most popular green composites studied in scientific literatures are reinforced with, e.g., flax, hemp, jute, and sisal fiber, and sustainable polymers such as polylactic acid (PLA), polyhydroxybutyrate (PHB), and poly(butylene succinate) (PBS) are typically used as the matrix [4]. Some examples of natural fiber-reinforced composites are PP reinforced with hemp, flax, and kenaf fibers [5], Supersap CLR matrix reinforced with sisal fibers [6], PLA/flax composites [7], PHB/piassava composites [8], and PLA/kenaf composites [9].

Experimental methods can be used to study new materials in the automotive industry. However, simulation tools based on the finite element method (FEM) as Abaqus, Ansys, and LS-DYNA have emerged as computer-aided engineering tools due to the elevated cost and time consumption of the experiments.

The usefulness of the FEM models to analyze the mechanical behavior of components made from biocomposites is unquestionable. In the past few years, the mechanical engineering field has been inseparable from such effective and powerful software. The theoretical principles behind it allow to for the development of new theories or criteria and ensure the accuracy of the simulation results, as Jalón et al. [10] did in their study about the impact behavior of flax fiber-reinforced polylactic acid (PLA) composite. They created the FEM model, simulated the impact scenario, and validated the constitutive model developed for natural fiber-based composites.

Moreover, FEM has also been further applied by many researchers in the new composite materials and automobiles field, in order to predict the failure mechanism, deformation behavior, and mechanical characteristics of automobile bumper beams made of natural fiber composites more effectively under various impact experiments. Hassan et al. [11,12] conducted their work on the low-speed impact study of an oil palm empty fruit bunch (OPEFB) fiber-reinforced composite bumper beam compared with a conventional aluminum one. Their numerical simulations (LS-DYNA) showed that the deformation was more severe in the natural fiber composite bumper than in the aluminum one, and 56% lighter. Therefore, they concluded that this biocomposite has the potential to replace aluminum in bumper beam applications. Arbintarson et al. [13] used an FEM model based on Solidworks software to study the behavior of a vehicle front bumper made of pineapple leaf fiber-reinforced composite in the front collision to a wall. They varied the impact velocity and concluded that the composite bumper could bear collision speeds up to 70 km/h. Ramasubbu and Madasamy [14] designed a sisal/kenaf fiber-reinforced-composite car bumper with a FEM model developed in Solidworks to simulate the drop-weight test. Their analysis of the NFRP bumper beam mechanical behavior was similar to a polycarbonate bumper. Rubio-López et al. [15] created a numerical FEM model in Abaqus/Explicit to predict the crashworthiness of all-cellulose composite (ACC) plates under low-velocity impact. After the simulation, the damage modes of the plate were given, and the energy threshold that led to its complete failure was obtained.

There are other studies in which researchers analyze the use of natural fiber-reinforced composites to enhance the chemical, functional, and mechanical performance of traditional composites such as those reinforced with glass fibers. Ghani et al. [16] studied the combination of glass/jute fiber-reinforced PBS composite material and the influence of the stacking sequence on the mechanical, thermal, and water-resistance properties of this composite. They found that the hybrid composite showed better mechanical performance than the pure composite, and they mentioned that the glass fiber layer on the surface could improve the impact performance of the combination. Murugu et al. [17] studied the quasistatic behavior of hybridized glass-/hemp-fiber-reinforced composite bumper using an FEM model developed in the Ansys. They concluded that this hybrid composite showed prominent properties to replace the conventional glass-fiber-reinforced composites.

Fiber-reinforced composites present distinct failure mechanisms to traditional metallic materials due to their anisotropy and heterogeneity. The different failure criteria in the literature consider three main failure modes to predict the failure of a fiber-reinforced composite: matrix cracking, fiber breakage, and delamination. However, many works have demonstrated that flax-fiber-reinforced composites are failed mainly by fiber breakage. Qian et al. [18] claimed that when the processing temperature exceeds a critical point, flax-fiber-reinforced composite fails during impacts, and the primary failure mode appears to be the fiber breakage of the material. Flax-fiber-reinforced composite thermally treated below 180 °C presents better fatigue and impact resistance. However, when the temperature exceeds this value, the mechanical properties of the composite are degraded, and the failure morphology will be changed. The work carried out by Díaz-Álvarez et al. [19] also reported that the fiber breakage occurs in the flax-fiber composite when impact energies are high, and the after-impact behavior of the structure presents promising results due to the absence of the delamination phenomenon. In addition, they also demonstrated that for low impact energies, the material fails due to matrix cracking.

Numerous studies show that the bumper beam geometry remarkably influences the structure impact behavior. Significantly, it depends on the structure curvature, cross-section geometry, and fiber orientation. For instance, in a previous work [20], four types of bumper beams that differed in the roundness of the cross section were studied under low-velocity impact test and bending after impact test. The results showed that the cross-section selection depends on whether the purpose is to achieve a better impact or postimpact performance. Additionally, the biocomposite-made bumper beams all show low peak force, which means that using green material can effectively reduce the acceleration during impact.

The main novelty of the present study is the development of an effective FEM model for predicting the damage behavior of bumper beams made entirely with fully-biodegradable flax-fiber-reinforced PLA composite. Four cross-sections were modeled to evaluate the prediction capacity of the model. First, tensile and peeling tests were carried out to obtain the composite mechanical properties and fracture toughness. Then, the properties were introduced in a numerical model developed to reproduce the low-velocity impact tests. Subsequently, the model was validated by comparing numerical predictions and experimental tests published in the previous work [20] in terms of absorbed energy and contact force history. Moreover, some specimens were inspected using X-ray scan tomography to validate the accuracy of the delamination extension predicted by the FEM model. Finally, the model results are used to analyze the different failure modes and energy absorption mechanisms as a function of the impact energy and cross-section roundness.

## 2. Methodology

The finite element method was used to create numerical models in Abaqus/Explicit for the mechanical response prediction of flax/PLA bumper beams under low-velocity impact tests. The experimental results needed for the numerical model calibration were published in previous work [20]. Four models were created to reproduce the geometry of the corresponding different bumper beams (named Type A–D). The methodology was divided into four steps:Definition of the numerical model. It includes an intralaminar failure model, which considers nonlinear viscoplastic behavior and the influence of strain rate, and an interlaminar failure model based on cohesive interaction.Most of the material parameters were obtained through experimental characterization tests. However, some parameters included in the cohesive interaction are difficult to obtain experimentally. Thus, they were fitted using the bumper beam Type A results.Once the cohesive parameters were fitted using only the experimental results of the bumper beam Type A, the experimental results of the other bumper beam types (B, C, and D) were used to validate the prediction accuracy of the numerical model.Finally, the validated numerical models are used to analyze the failure modes and the influence of the cross section on the absorbed energy.

### 2.1. Numerical Model

Before the impact test, every specimen’s weight and dimension was measured for the modeling in Abaqus/Explicit. The density of the flax/PLA composite was equal to 1.14 ton/m^3^. According to the previous studies of flax/PLA, this biocomposite presents nonlinear elastic viscoplastic behavior. Therefore, the traditional mechanical behavior model for composites considering linear elastic behavior up to failure cannot be used to reproduce the behavior of green composites. In the present model, two regions were considered in the stress–strain curve: elastic and viscoplastic. As shown in Figure 1, the first elastic stage is assimilated to linear elastic with mechanical properties of E = 5 GPa and Poisson ratio = 0.3; the second region is viscoplastic and was defined using experimental strain-rate-dependent data. Tensile tests were carried out in the laboratory at different velocities to characterize the influence of strain rate. A constitutive viscoplastic model was defined in a previous work [21], which was implemented in an FEM model to reproduce the mechanical behavior of flax/PLA plates either under low-velocity impacts [10] or machining operations scenarios [22,23].

Finally, a VUSDFLD user subroutine was used to define an ultimate strain criterion to remove elements that reach the ultimate strain. Figure 2 shows the function of the ultimate strain as a function of the strain rate implemented in this subroutine. For strain rates greater than 0.0154 s^−1^, the ultimate strain was fixed as 0.056; otherwise, the ultimate strain obeys a linear function. These values are based on the results obtained in the experimental tensile tests; more detail can be found in [21]. Therefore, the subroutine calculates the ultimate strain as a function of the strain rate and deletes elements that exceed the ultimate strain.

The model developed in Abaqus/Explicit is composed of three solids to mimic the actual impact scenario: striker, supports, and bumper beam, as shown in Figure 3, to reproduce the low-velocity impact tests performed [20]. The experimental tests were performed in three-point bending configuration set in a drop-weight tower to analyze the impact behavior of four geometrically different flax/PLA bumper beams. An INSTRON-CEAST Fractovist 6785 drop-weight tower (High Wycombe, Buckinghamshire, UK) was used; the instrumented striker that moved along the vertical guide rail had customized features: Charpy nose, the weight of 5.93 kg and diameter of 20 mm. 

The striker was modeled as a 3D discrete rigid shell, element type R3D4, U-form, with overall length of 70 mm, overall height 25 mm, and radius of the bottom surface of 10 mm. It has a point mass of 5.93 Kg applied in the up-center point of the structure, called the reference point (RP), see Figure 3), where a multipoint constraint (MPC) was created to tie all the nodes of the striker to the RP, constraining the motion of the slave nodes to the motion of a single point. This simplification was assumed to reduce the computational cost because the stiffness of the steel striker was much higher than the bumper beam. An additional advantage of this simplification is that the striker results (velocity, acceleration, reaction forces…) can be registered in a single node. All the degrees of freedom of the striker nodes were restricted except the vertical displacement. Moreover, a predefined velocity field was defined with values between 1.006 m/s to 4.859 m/s (equivalent to 3–70 J of kinetic energy).

The supports are composed of two cylinders that hold the bumper beam and four pins that limit its lateral movement. Since no permanent deformations after impact were found, the supportive structures were modeled as homogeneous solids with linear elastic properties of ordinary steels (E = 210 GPa and Poisson Ratio = 0.3), and the element type is C3D8R. In addition, their boundary conditions were *Encastre*, see Figure 3. 

The actual beam is modeled as a four-layer 3D deformable body, including cohesive contact properties between layers to reproduce delamination failure. All four bumper beam types were modeled having the same average thickness (t = 2.5 mm), width (w = 106 mm), and length (L = 210 mm). However, they differ in the cross-section geometry, as depicted in Figure 4. Their fillet radius increases from Type A to Type D according to the bumper beam geometries used in the experimental tests of a previous study [20]. The maximum radius corresponds to a semicircular geometry and the minimum radius is determined by the compression-molding manufacturing method. The characteristics of the four cross sections are summarized in Table 1.

The FEM model was verified using the experimental results of low-velocity impacts. The model mesh should be sufficiently refined to ensure that the results are reliable and accurate, considering that the computational cost rises according to the model refinement level. Thus, it was essential to find the optimal mesh to acquire a balance between the accuracy of the results and the computational cost. 

Two variables: the absorbed energy (Eabs) for the whole model and the peak force (Fmax), were compared with the experimental values to verify the model prediction capacity. Three meshes (coarse, fine, and very fine mesh) were chosen to analyze the convergence of results for an impact energy of 60 J. The three meshes included a nonuniform distribution of elements along the beam length to reduce the processing time without compromising simulation accuracy. The double-bias seeding changed the mesh density progressively from the coarsest end to the finest center area of the bumper beam, emphasizing the reliability of the impact zone; see Figure 3.

For a very fine mesh (101,000 elements), the model lasted 18 h to complete the simulation and the predicted results were 57.66 J of absorbed energy and 2466.6 kN of peak force. On the other hand, for the fine mesh (65,000 elements), the simulation lasted 11 h and the results were similar: 58.84 J of absorbed energy and 2490.1 kN of peak force. Therefore, the fine mesh of approximately 65,000 elements was selected with sizes from 4 mm in beam ends to 1 mm in the contact area. 

### 2.2. Fitting

The fitting process was conducted on the type A bumper beam, while other bumper beam types were used in the validation process. As mentioned in the previous section, cohesive interaction was used to simulate the interlaminar damage onset and propagation. Polylactic acid (PLA) thermoplastic was used to bind the different layers of the bumper beam, working as the cohesive element for the entire system; thus, its mechanical properties were taken as references for the cohesive properties in Abaqus/Explicit (see Table 2).

As the interface thickness of the bonded layers was negligible, a cohesive constitutive response based on traction–separation law was used for the numerical model. Abaqus offers the traction–separation model that assumes an initial linear elastic behavior followed by two stages, the initiation of damage (the onset of degradation) and the evolution of damage (the propagation of the damage up to failure). The elastic behavior is defined as an elastic matrix that relates the nominal and shear stress to the nominal and shear separation [25]. Therefore, the stiffness coefficients (*K_nn_* = *K_ss_* = *K_tt_*) were specified as the Young modulus of PLA for uncoupled traction–separation behavior (Table 2). These values were not modified during the fitting process.

To define the damage initiation stage, the maximum nominal stress criterion was considered, i.e., when the maximum contact stress ratio reaches one, as shown in Equation (1), the damage of the instances will initiate.
(1)$$max\left\{\frac{t_{n}}{t_{n}^{o}},\frac{t_{s}}{t_{s}^{o}},\frac{t_{t}}{t_{t}^{o}}\right\}=1$$
where $t_{n}$, $t_{s}$, and $t_{t}$ are the normal and the two shear tractions from the nominal traction stress vector, respectively.

The peak values of the contact stress ($t_{n}^{o},\;t_{s}^{o},\;\;t_{t}^{o}$) were established during the fitting process. The initial value for these three parameters was equal to the PLA tensile strength, 54.27 MPa (Table 2). However, low absorbed energy and poor delamination response were generally detected in the simulations. The comparison with experimental results showed that these parameters required modification to increase the model accuracy. By the trial-and-error method, lower values seemed to give more reasonable results of energy and force, so the final attempt led to final stress of $t_{n}^{o}=t_{s}^{o}=t_{t}^{o}=30\;\mathrm{MPa}$.

For the damage evolution specification, the approach used in this study was based on the dissipated energy due to failure (fracture energy), which is equal to the area under the traction–separation curve, and a linear softening behavior was considered to simplify the scenario. Once that, Benzeggagh–Kenane (BK) fracture criterion was chosen to compute the damage evolution of the cohesive interface, and the formulation is given by
(2)$$G_{IC}+\left(G_{IIC}-G_{IC}\right){\left\{\frac{G_{S}}{G_{T}}\right\}}^{\eta }=G_{C}$$
where $G_{IIC}=G_{IIIC}$, $G_{S}=G_{IIC}+G_{IIIC}$, $G_{T}=G_{IC}+G_{S}$, and η is an empiric value called cohesive property parameter. According to Riccio et al. [26], the cohesive property parameter comes from experimental tests, ranging between 1 and 1.6. In order to simplify the study, η was equal to 1 for the numerical model.

Since green composites are still in their early stages of development, these materials are not commonly tested through the double cantilever beam test method (DCB) to determine the interlaminar fracture toughness energy (*G_Ic_*), and few experimental tests can be found in the literature. To obtain the interlaminar fracture toughness energy in Mode I (*G_Ic_*), double cantilever beam (DCB) tests according to the ASTM D5528-13 standard [27] were carried out, Figure 5a). It should be noticed that this standard was developed for carbon-fiber-reinforced plastics. However, it was used in this study because no specific standard for natural fiber-reinforced composites was found. According to the ASTM D 5528 standard, *G_ic_* can be found using the area of the force–displacement curve, Figure 5b), using Equation (3):(3)$$G_{Ic}=\frac{A}{a\cdot w}$$
where *A* represents the area below the load–displacement diagram, *a* indicates the propagated crack length, and *w* is the width of the specimen, Figure 5c).

The *G_Ic_* value obtained in the experimental tests conducted on flax/PLA specimens was 2 mJ/mm^2^. However, no experimental tests were conducted on the flax/PLA specimens to determine the fracture toughness energy mode II (*G_IIc_*) and mode III (*G_IIIc_*). The reason is the lack of specific standards and the difficulty of applying standards developed for CFRPs because the stiffness of natural fibers is much lower than that of carbon fibers. Therefore, to fit *G_IIc_* and *G_IIIc_*, values between 2 and 3 mJ/mm^2^ were checked. This range was selected because the fracture toughness energy modes II and III are equal to or higher than mode I in composites. After the fitting process, values of fracture toughness energies were fixed at *G_Ic_* = *G_IIc_* = *G_IIIc_* = 2 mJ/mm^2^.

As a result of the fitting process, the model reasonably predicted the contact force history, the evolution of absorbed energy, and the delamination extension for different impact energies on bumper beam Type A.

#### 2.2.1. Contact Force History

Figure 6 compares the experimental results and the numerical predictions regarding the contact-force history for impact energies equal to 30 J, 50 J, and 70 J. These impact energies were selected because they were representative of the three typical impacts: damage located in contact area at 30 J, complete failure of the bumper beam at 70 J, and intermediate case at 50 J. It can be seen that the model can predict not only the peak forces but also the different trends of contact force history and the duration of the impact event. The FEM model can predict the force history recorded in experimental results for 30 J and 70 J with a great accuracy; however, the accuracy in the 50 J impact test is lower. There are two possible reasons for this lack of precision: first, there is an intrinsic scattering in experimental results for composites reinforced with natural fibers; and second, 70 J is a case of high impact energy that produces the total failure of the beam, and 30 J is a case of low impact energy with localized damage, while 50 J is an intermediate impact energy, and the transition energies are more difficult to reproduce with numerical of theoretical models.

#### 2.2.2. Absorbed Energy

The evolution of the absorbed energy is shown in Figure 7. For impact energies of 30 J or 50 J, the curves can be divided into two stages. First, absorbed energy increases until it reaches the maximum value, the impact energy. Then, the absorbed energy decreases until the final value. The decrease in the second stage is produced because part of the absorbed energy was elastic energy that the striker recovered in the form of kinetic energy. 

On the other hand, for impact energy of 70 J, the absorbed energy increases until a maximum value of around 60 J because that is the maximum capacity of energy absorption of the bumper beam. When that energy is achieved, a complete failure is produced in the bumper beam, and the contact forces drop to zero, see Figure 6. The model can reproduce the absorbed energy evolution and predict the bumper beam total absorbed energy.

#### 2.2.3. Delamination Damage Extension

Finally, three Type A specimens were subjected to X-ray scan tomography to evaluate the damage after impact. The equipment used—a Phoenix v/tomex of GE Sensing and Inspection Technologies X Ray company—has an x-ray tube with a 140 kV nanofocus. Figure 8, Figure 9 and Figure 10 show a comparison between the tomography results and the numerical model predictions. At impact energy of 30 J, Figure 8, the predicted extension of the delamination in the top corner of the middle section (section B) is quite similar to the experimental scenario, which indicates a good accuracy of this FEM model. The numerical model also predicted slight delamination at the bottom corners of the sections near the supports (sections A and C). However, these delaminations were not detected in the tomography inspections; this discrepancy can be explained by the boundary conditions that cannot reproduce the complex nature of experimental impact test. The delaminations seem to occur only at the impact zone of the bumper beam.

When impact energy increases to 50 J (Figure 9), the damage in the central section also increases considerably. Delaminations can be found at the top corners and on the web. Moreover, a permanent deflection can be appreciated in the top flange. The model can reproduce this increase in delamination and permanent deformations due to plastic strains. For cross sections near the supports, both experimental and numerical results agree on the presence of small delaminations in the web and the bottom corners. The rest of the bumper beam sections do not present delaminations.

If impact energy increases to 60 J (see Figure 10), the numerical model again accurately reproduces the damage in the impact section of the structure. Moreover, smaller delaminations appear in the top flange and the web of the cross sections near the support. Finally, considering the numerical model prediction capability in terms of contact force, absorbed energy, and damage extension, the fitted parameters were fixed, and the results in bumper beam types B, C, and D were used to validate the numerical model.

## 3. Model Validation: Comparison with Experimental Data

After the fitting process, the same loading and boundary conditions were used for the remaining types of bumper beams by only changing the cross-section geometry. Validation of the models is studied in this section by comparing numerical and experimental results.

Figure 11 shows the absorbed energy versus the impact energy curves (Eabs-Eimp) of the four bumper beams in which the numerical prediction is compared with the experimental data. Generally, the numerical results show excellent agreement with the experimental data, except for 30–50 J. These disagreement produces different slopes in numerical and experimental curves. The main reason for this underestimation of absorbed energy is that the threshold energy of the bumper beams occurs within 30–50 J. For impact energies near the threshold energy, the behavior is intermediate between local damage in the contact area and complete failure of the bumper beam. This intermediate behavior is the most difficult to reproduce with a numerical model. Moreover, as massive damage (matrix cracking, fiber breaking, and delamination) initiates at this range, more uncertainty makes the scenario harder to predict and explains the difference between the experimental and numerical values. Nevertheless, the numerical model can reproduce the tendency and the values of absorbed energy with reasonable accuracy for the four bumper beam types.

Figure 12 shows the contact force versus displacement curves for all the bumper beam types. Three impact energies were selected because they are representative of the three different impact behaviors. The lower impact energy, 30 J, represents a typical low-velocity impact with localized damage in the contact area. The force increases with displacement until the peak force is reached, then both force and displacement decrease until force drops to zero, but there is a permanent displacement around 10–15 mm. The impact energy of 70 J is the highest impact energy analyzed in this work and represents the complete failure of the bumper beam. The force increases with displacement until the peak force is reached and then drops to zero, but the displacements always increase, indicating that the striker does not rebound. The impact energy of 50 J represents an intermediate behavior with significant damage in the bumper beam but without a complete failure. The force–displacement curve at this energy is also a transitional curve between low and high impact energy. The force increases with displacement until peak force is reached, then the force decreases, but the displacement keeps increasing until maximum displacement is reached; finally, both force and displacement decrease. It can be observed that the numerical model can predict the experimental trends for the three impact energies, even the unique case of Type C at 50 J (Figure 12c), where there is a second peak force. This unusual appearance might point out that the cracks start to appear in the beam structure, and this tendency grows more remarkable from Type A (Figure 12a) to Type D (Figure 12d). Therefore, the numerical model showed an excellent prediction capability in terms of contact force, displacement, and energy absorption. 

## 4. Damage Analysis

The validated numerical model was used to analyze the damage evolution and better understand the energy-absorption mechanisms of the biocomposite bumper beams. Figure 13, Figure 14, Figure 15 and Figure 16 show the delamination pattern in the middle cross section for all the bumper beam types at different impact energies. The goal of the analysis of damage in the middle cross section is the explanation of the results shown in Figure 11. The energy-absorption capability of the bumper beam Type A is not reached because the absorbed energy at 70 J is clearly higher than that at 60 J. In bumper beam types B and C, the absorbed energy at 60 J is almost equal to at 70 J, indicating that 60 J is the threshold energy where the maximum energy absorption capability is reached. This maximum energy-absorption capability was reached at 50 J for bumper beam type D because the absorbed energy does not increase more at 60 J or 70 J. 

The evolution of delamination with impact energy in bumper beam Type A can be observed in Figure 13. For the impact energy of 30 J (Figure 13a), delamination is localized at the top corners, but the center of the top flange and the webs are free of delaminations. When impact energy increases to 40 J and 50 J (Figure 13b,c), delaminations propagate along the top flange and the webs but do not entirely fail. It can also be seen that the cracking of the section begins from the corner at 40 J and 50 J, which corresponds exactly to the anomaly that appears in the force-displacement at 50 J of Figure 12. When impact energy is equal to 60 J (Figure 13d), delamination is propagated through the top flange, and still, part of the webs is free of delamination. Only at the impact energy of 70 J, (Figure 13e), all the webs and top flange are delaminated; thus, the stiffness of the cross section drops to zero, and the absorbed energy capability of the bumper beam Type A is reached. This tendency agrees with the results shown in Figure 11, indicating that absorbed energy increases with impact energy in the range of impact energy studied in this work.

Figure 14 shows the delamination of bumper beam Type B for different impact energies at the middle cross section. For impact energy of 30 J (Figure 14a), delaminations are located at the top corners. When impact energy increases to 40 J (Figure 14b), there is a slight propagation of delaminations, but they are located around the top corners. If impact energy increases to 50 J or 60 J (Figure 14c,d), delaminations propagate in the top flange and the webs, but some parts of the webs and top flange are still free of delamination. For impact energy of 70 J (Figure 14e), delaminations are extended through the whole top flange and webs; thus, the stiffness of the cross section is lost, and the energy-absorption capability of the bumper beam Type B is reached. Figure 11 shows that the absorbed energy of bumper beam Type B increases with impact energies up to 70 J. However, the increment from 60–70 J is almost negligible, indicating that the threshold energy is probably between 60 J and 70 J.

The predictions of delaminations in the bumper beam Type C under different impact energies at the middle cross-section are shown in Figure 15. Delaminations are located at the top corner for the impact energy equal to 30 J (Figure 15a). For impact energies of 40 J and 50 J (Figure 15b,c), delaminations propagate through the top flange and webs, but with some parts of them free of delaminations. Furthermore, for impact energies of 60 J and 70 J (Figure 15d,e), delaminations are entirely extended in the top flange and webs, indicating that the cross-section stiffness drops to zero and the maximum energy absorption capability of the bumper Type C is reached at 60 J. These results agree with those shown in Figure 11 because the absorbed energy of bumper beam Type C increases with impact energy until 60 J, and the absorbed energy at 70 J is almost equal to that of 60 J.

Figure 16 shows the bumper beam Type D delamination at the middle cross section for different impact energies. For impact energy of 30 J (Figure 16a), delaminations are concentrated at the ±45° angles of the arc. If the impact energy increases to 40 J (Figure 16b), delaminations propagate toward the arc center, but the arc bottom part is free of interlaminar damage. For impact energies equal to or higher than 50 J (Figure 16c), the whole arc of the cross section is delaminated; thus, the cross section loses its stiffness, and the energy-absorption capability of the bumper beam Type D is reached. The results shown in Figure 11 confirm that 50 J is the impact energy threshold because the absorbed energy increases with impact energy until 50 J, and the absorbed energy at 60 J and 70 J is almost equal to that of 50 J. 

The comparison of the different bumper beam types indicates that the lower the roundness of the cross section, the higher the energy-absorption capability. The squarest cross section, Type A, is the bumper beam with the highest energy absorption capability. On the other hand, the hemi-circumference cross-section-shape bumper beam, Type D, with the highest roundness, presents the lowest energy-absorption capability. Finally, the energy absorbed by types B and C is intermediate because their roundness is also transitional between types A and D. The main reason for this phenomenon is that the moment of inertia of the cross section decreases with the roundness. Therefore, contact forces also decrease with the cross-section roundness, as shown in Figure 12. The absorbed energy increases with the contact force; thus, bumper beam Type A shows the highest energy-absorption capability, and Type D the lowest.

Another reason behind this phenomenon is that the threshold energy is associated with the damage extended to the whole cross section. The delamination onset is located at the cross-section corners. If the corner radius is small, as in Type A, delaminations are more concentrated around the corners. In contrast, a high radius implies that delaminations are more prone to be distributed through the entire cross section. In other words, delamination can propagate prematurely through the cross section at lower impact energies. 

These results are in clear contradiction with previous works on impact on flat plates of NFRPs [18,19], where the main failure mode observed is fiber breakage, and delaminations are almost negligible. This work, and a previous study on bumper beams manufactured with green composites [20], demonstrate that the impact behavior of curved specimens as bumper beams are strongly dominated by delaminations, being the delamination effect negligible only on flat plates.

To obtain a better understanding of the differences between the impact behavior of the four cross sections, a comparison of the failure modes at 60 J was performed using a high-speed camera to record the impact tests. Figure 17 compares the bumper beam deformed shape predicted by the numerical model with the experimental results at the impact energy of 60 J. It can be seen in Figure 17a) that types A and B do not fracture at 60 J. The bumper beam absorbed most of the kinetic energy, and the rest was returned to the striker with the elastic recovery of the bumper. Therefore, as the impact performance of bumper beam types A and B has not reached the limit, they can still have residual properties in postimpact testing. However, bumper beams C and D have already experienced a complete failure at this impact energy.

## 5. Conclusions

The numerical model developed in Abaqus/Explicit shows accurate predictions on the damage behavior of the four types of bumper beams made of flax/PLA green composites. Its validation was carried out by comparing with the experimental low-velocity impact results: the energy absorption, contact-force history, and extension of delamination. Moreover, the model was also used to analyze the damage evolution and the energy-absorption mechanisms of the biocomposite bumper beams as a function of the impact energy and cross-section roundness. The reliability of the numerical model in predicting the growth of the delamination phenomenon was verified by comparing the damage with the X-ray scan tomography of Type A. 

From the study, several conclusions are drawn:-Regarding the prediction of the absorbed energy of the bumper beam, the numerical results show excellent agreement with the experimental data. The model is able to predict the different behavior as a function of impact energy, from localized damage to complete failure.-Analysis of the force–displacement curves at impact energies of 30 J, 50 J, and 70 J shows that the permanent deformations of the bumper beam initiate at 30 J; the damage on the structure becomes significant and shows different severity for the four types at 50 J; and the complete failure of the beams is produced at 70 J.-Experimental and numerical results revealed that the threshold energy, where the maximum energy absorption capability is reached, for Type A is over 60 J; for Type B and C is around 60 J; and for Type D is at 50 J.-The damage evolution showed that the delamination manifests initially in the section corners and then spreads further. It implies that delaminations are more prone to propagate through the entire cross section and more prematurely for rounder types.-Adding the fact that the rounder section presents smaller peak force and threshold energy. Therefore, the squarest cross section, Type A, is the bumper beam with the highest energy-absorption capability. Conversely, the roundest one, Type D, presents the lowest energy-absorption capability. The lower the roundness of the cross section, the higher the energy-absorption capability.-Lastly, the numerical model predicted the same deformed shape of the four bumper beams under 60 J as the experimental scenario.

The development of a numerical model that can accurately predict the impact behavior of structures manufactured with green composites opens new lines of research in the use of green composites in structures designed to absorb energy from impacts. However, to facilitate the use of green composites in industry, further research must be focused on the impact behavior of green composites after ageing tests.

## Figures and Tables

**Figure 1 materials-15-05480-f001:**
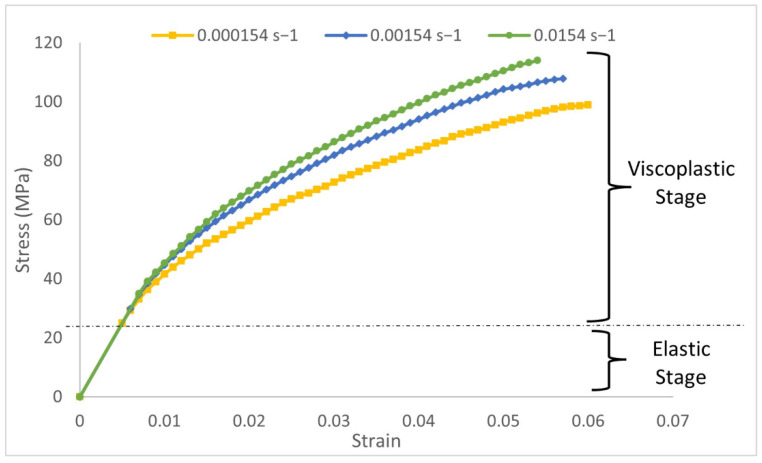
Stress-strain curves of flax/PLA biocomposite obtained in tensile tests at different strain rates.

**Figure 2 materials-15-05480-f002:**
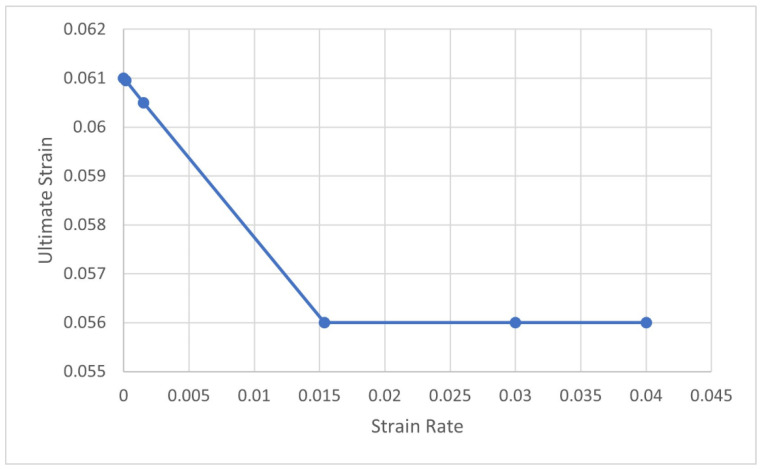
The ultimate strain as a function of the strain rate. The element-deletion criterion is implemented in the VUSDFLD subroutine.

**Figure 3 materials-15-05480-f003:**
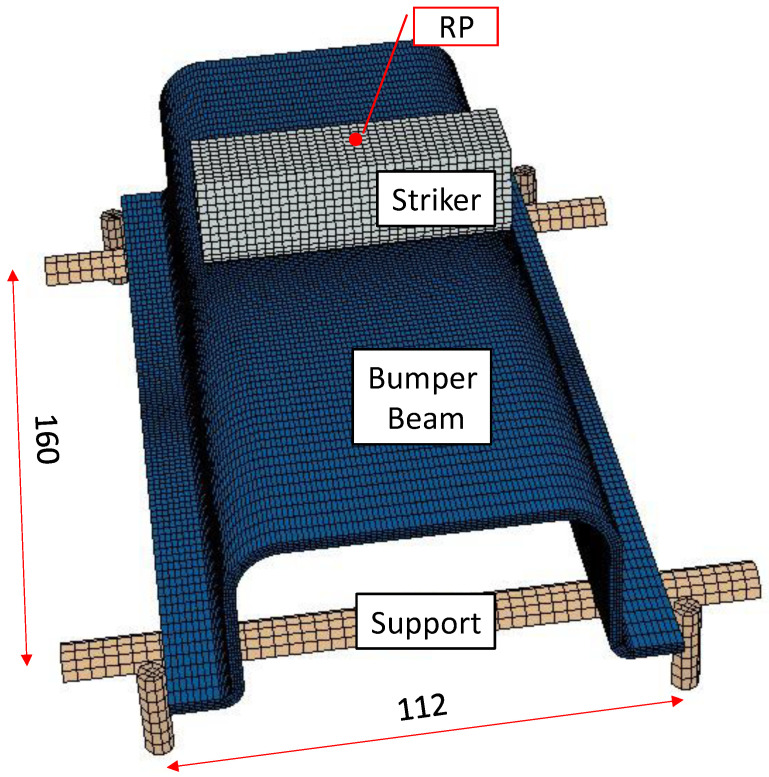
Numerical model of the bumper beam under low-velocity impact test. The overall setting and dimension.

**Figure 4 materials-15-05480-f004:**
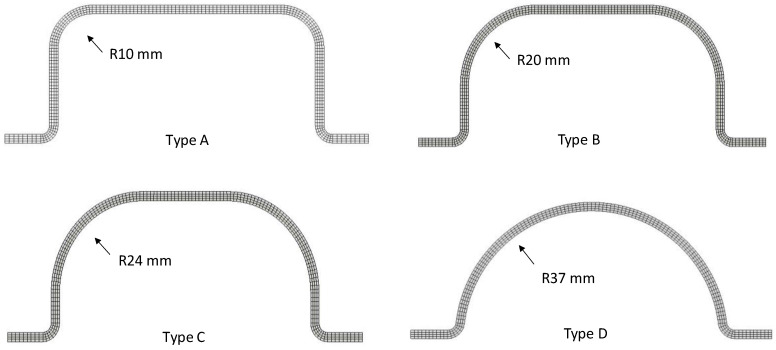
The cross-section geometry of the bumper beams.

**Figure 5 materials-15-05480-f005:**
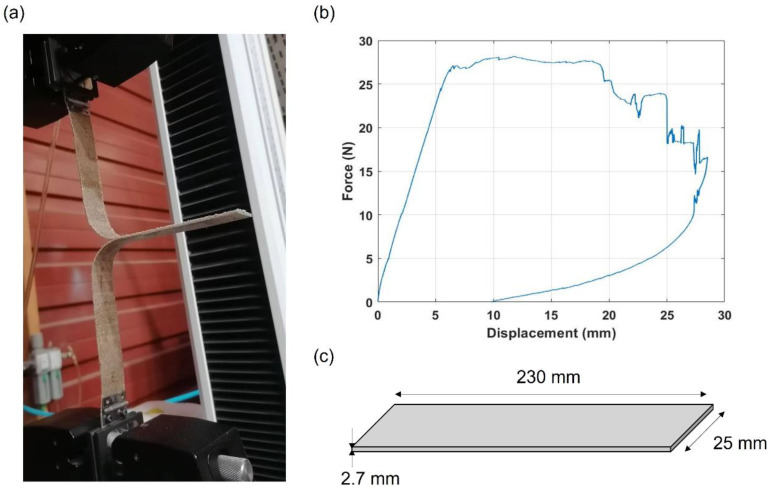
Experimental tests conducted to determine the interlaminar fracture toughness energy in Mode I. (**a**) Experimental setup; (**b**) load–displacement curve; (**c**) geometry of the specimens. Dimension not to scale.

**Figure 6 materials-15-05480-f006:**
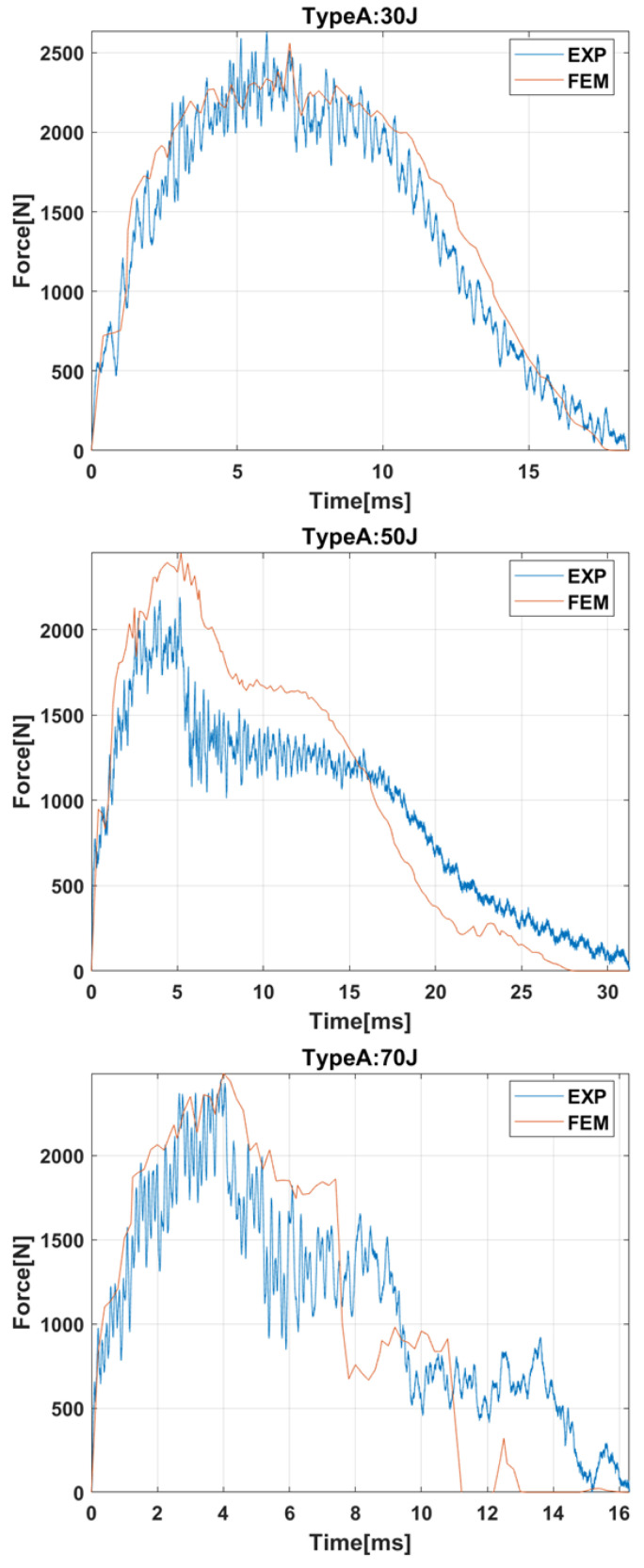
Force history of bumper beam type A under impact energies of 30 J, 50 J, and 70 J.

**Figure 7 materials-15-05480-f007:**
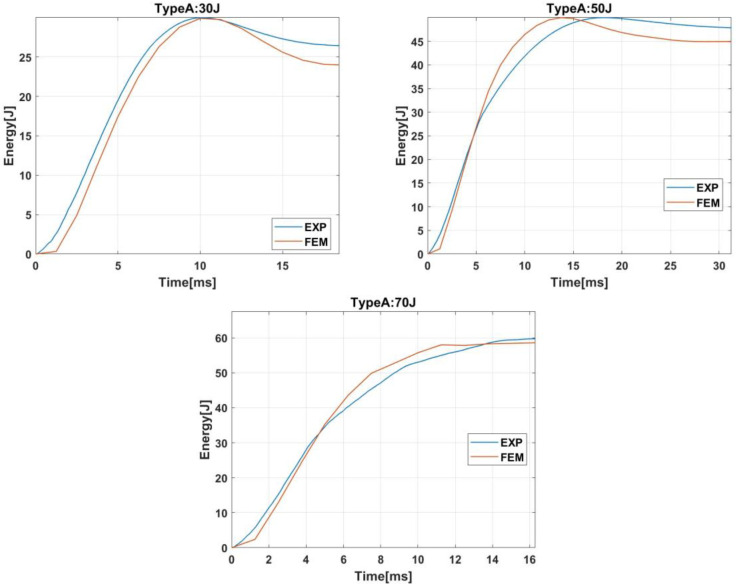
Energy history of bumper beam type A under impact energies of 30 J, 50 J, and 70 J.

**Figure 8 materials-15-05480-f008:**
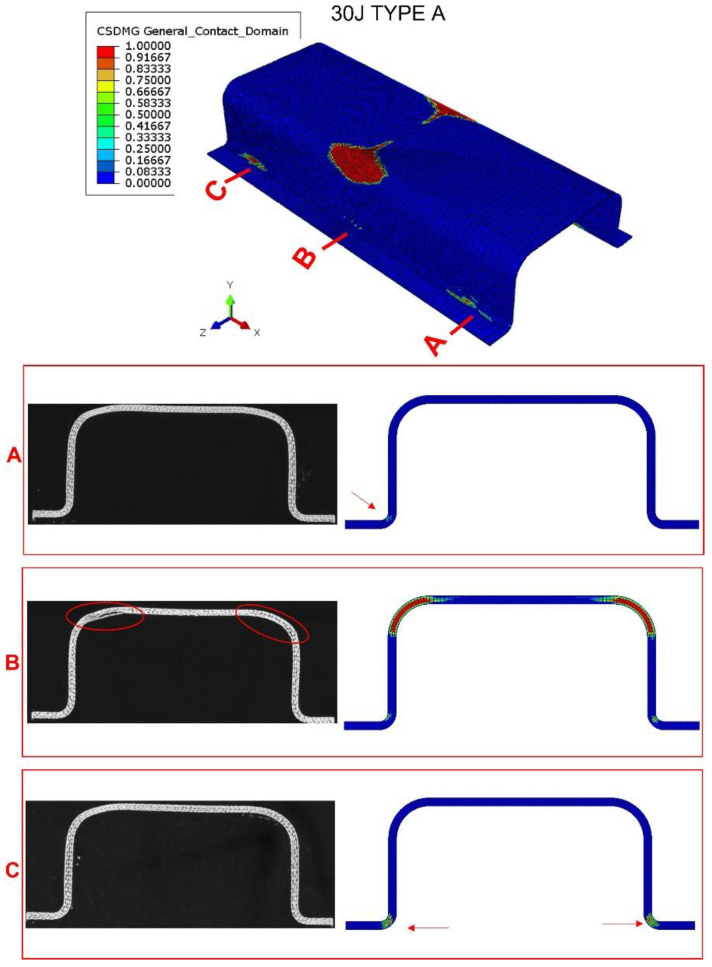
Tomography of bumper beam type A, impact energy of 30 J; and comparison with numerical results. Observation of the interlaminar failure in the ends of the beam (sections A and C) and in the impacted section (section B). Red arrows point the appearance of failure in sections A and C in the numerical screenshot, red ovals point to failure at fillet radius in section B.

**Figure 9 materials-15-05480-f009:**
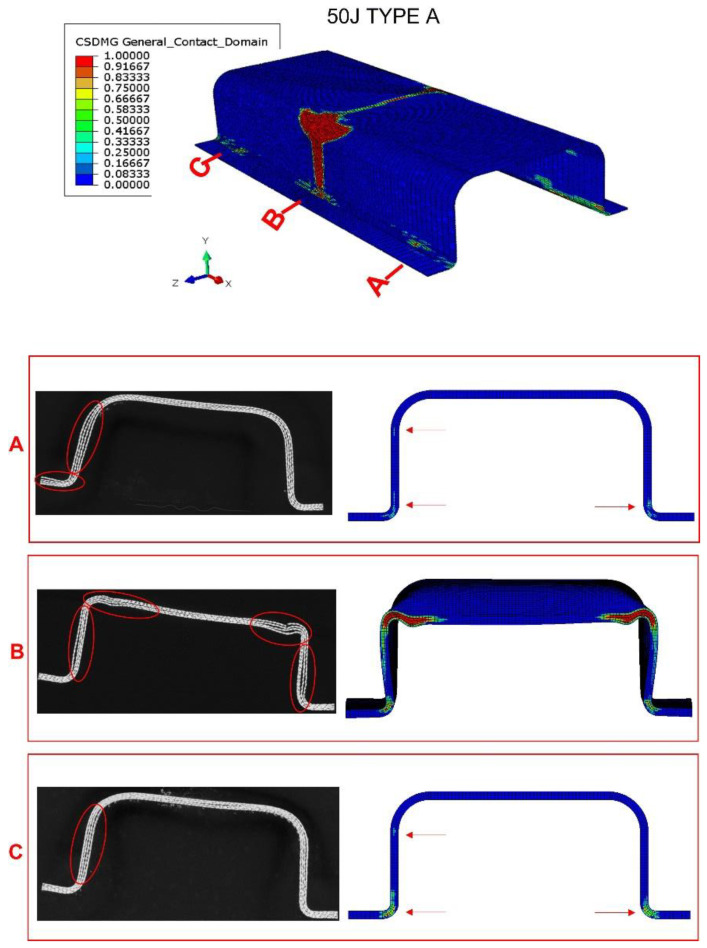
Tomography of bumper beam type A, impact energy of 50 J; comparison with numerical results. Observation of the interlaminar failure in the ends of the beam (sections A and C) and the center of impact (section B). The red arrows in the numerical screenshot point the appearance of failure in sections A and C.

**Figure 10 materials-15-05480-f010:**
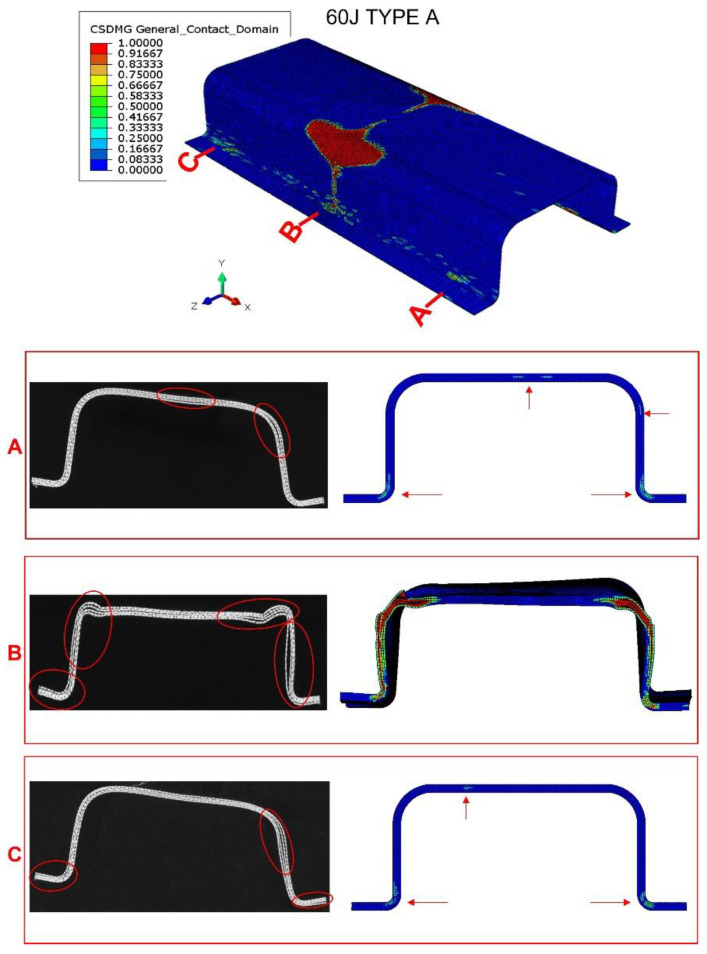
Tomography of bumper beam type A, impact energy of 60 J; comparison with numerical results. Observation of the interlaminar failure in the ends of the beam (sections A and C) and the center of impact (section B). The red arrows in the numerical screenshot point the appearance of failure in sections A and C.

**Figure 11 materials-15-05480-f011:**
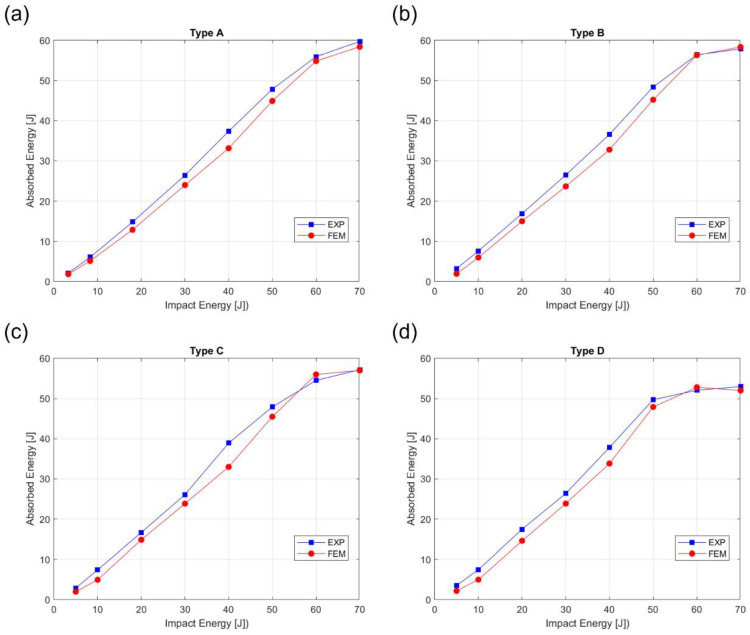
Absorbed energy as a function of impact energy. Experimental and numerical results comparison for all the bumper beam types. (**a**) E_abs_-E_imp_ for Type A; (**b**) E_abs_-E_imp_ for Type B; (**c**) E_abs_-E_imp_ for Type C; and (**d**) E_abs_-E_imp_ for Type D.

**Figure 12 materials-15-05480-f012:**
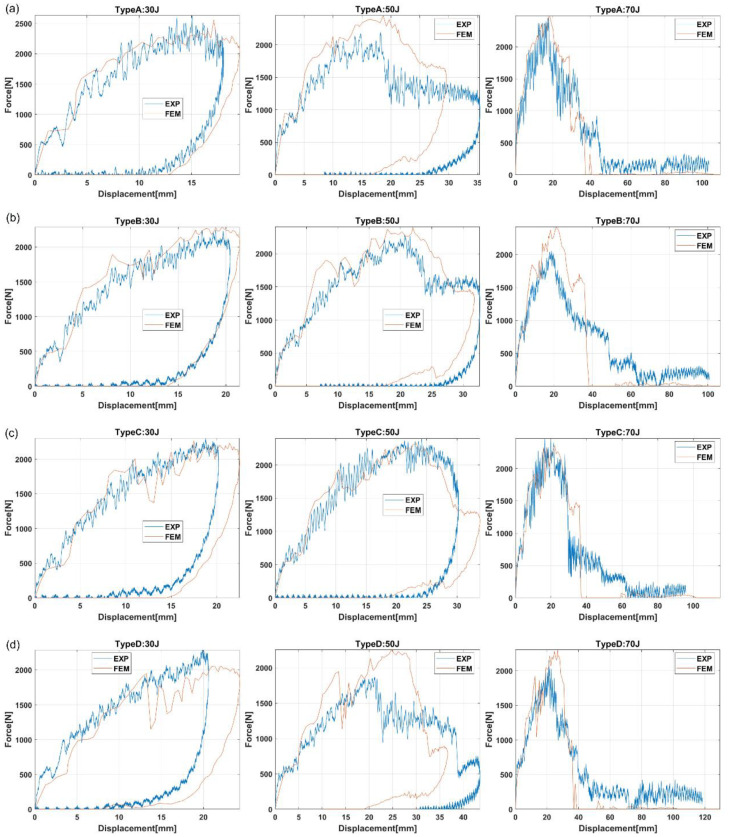
Force versus displacement curves of all the bumper beam types. Comparison between the experimental and the numerical results. (**a**) Curves for Type A; (**b**) Curves for Type B; (**c**) Curves for Type C; and (**d**) Curves for Type D.

**Figure 13 materials-15-05480-f013:**
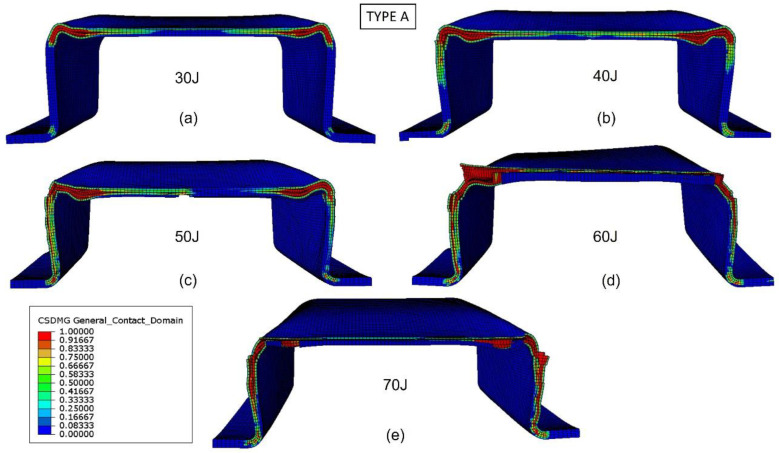
Delamination initiation criterion at the middle cross section of bumper beam Type A. Impact energy: 30 J (**a**), 40 J (**b**), 50 J (**c**), 60 J (**d**), and 70 J (**e**).

**Figure 14 materials-15-05480-f014:**
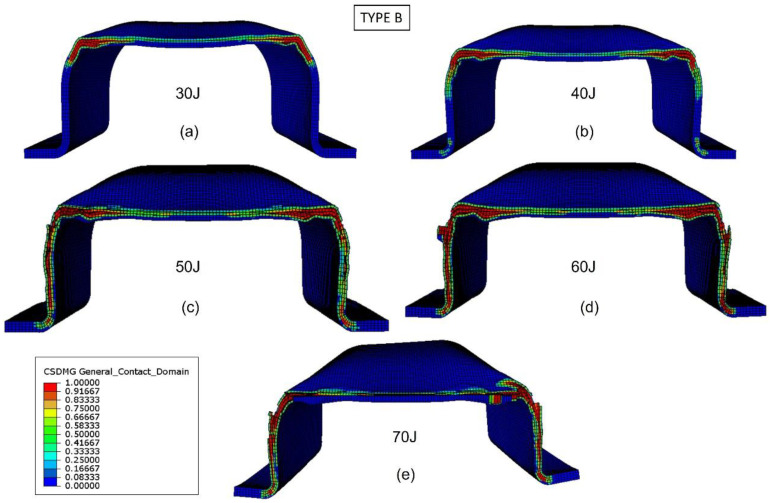
Delamination initiation criterion at the middle cross section of bumper beam Type B. Impact energy: 30 J (**a**), 40 J (**b**), 50 J (**c**), 60 J (**d**), and 70 J (**e**).

**Figure 15 materials-15-05480-f015:**
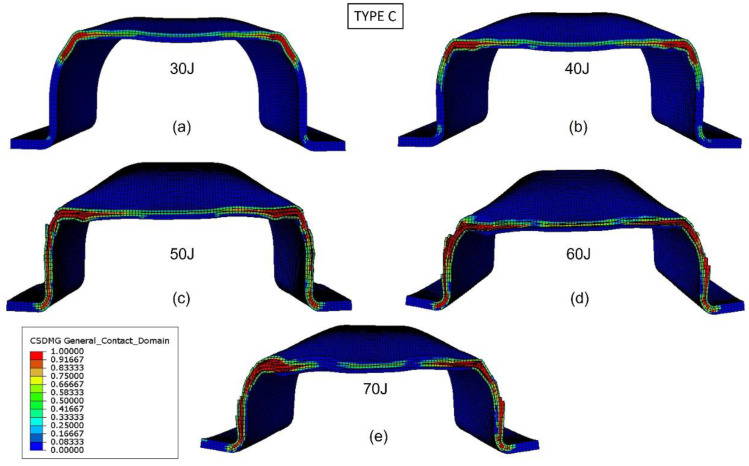
Delamination initiation criterion at the middle cross section of bumper beam Type C. Impact energy: 30 J (**a**), 40 J (**b**), 50 J (**c**), 60 J (**d**), and 70 J (**e**).

**Figure 16 materials-15-05480-f016:**
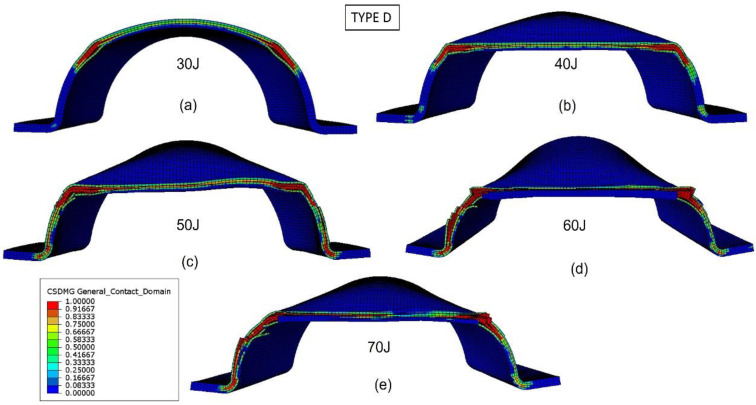
Delamination initiation criterion at the middle cross section of bumper beam Type D. Impact energy: 30 J (**a**), 40 J (**b**), 50 J (**c**), 60 J (**d**), and 70 J (**e**).

**Figure 17 materials-15-05480-f017:**
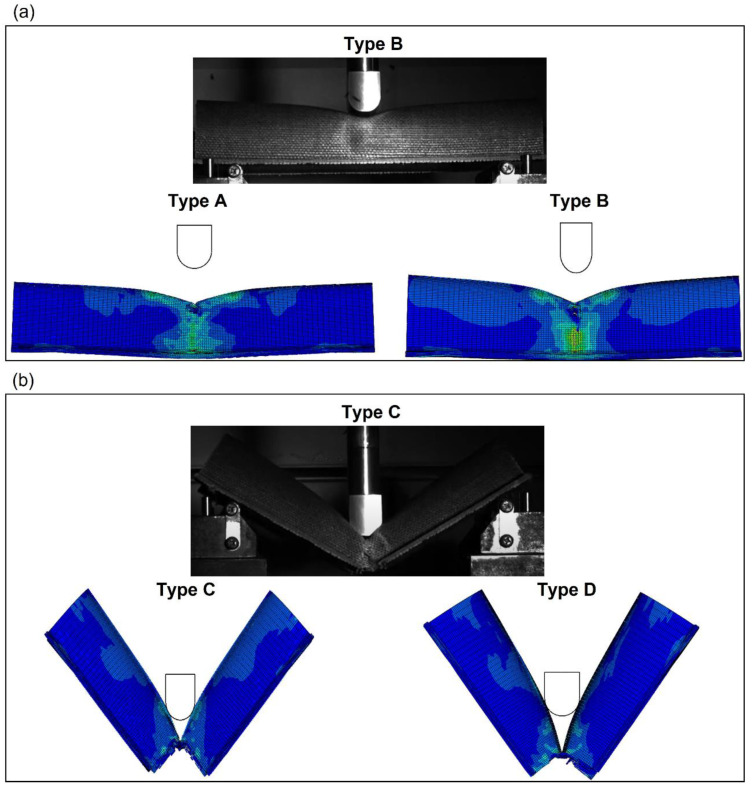
Side view of the bumper beams impacted at 60 J. (**a**) Photograph of Type B and numerical screenshots of Type A and B; (**b**) photograph of Type C and numerical screenshots of Type C and D.

**Table 1 materials-15-05480-t001:** Main geometric variables of the four bumper beam cross sections.

Bumper Beam	Radius (mm)	Height (mm)	Weight (g)
Type A	10	40	97.4
Type B	20	40	96.8
Type C	24	40	95.7
Type D	37	40	84.6

**Table 2 materials-15-05480-t002:** Mechanical properties of PLA, data from [24].

Mechanical Properties	PLA
Tensile strength (MPa)	54.27
Young modulus (MPa)	3180

## Data Availability

The data presented in this study are available on request from the corresponding author. The data are not publicly available due to privacy.

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
