# Peer review of "On the Numerical Modeling of Flax/PLA Bumper Beams"

_materials, 2022, doi:10.3390/ma15165480_

Round 1

Reviewer 1 Report

On the numerical modelling of flax/PLA bumper beams.

The article is well written. A few comments are given below;

The abstract could be revised with quantitative results.

There is a need to update the introduction with some latest studies.

The different numerical models could be summarized in the table along with their variables.

Figure 5b axis are not clear.

Also figure 6 is not readable.

Section 2.2 could be rearranged with subheadings for better understanding.

Figure 11, stiffness or slop is quite different, the authors must discuss this issue.

Colors are shown in the figure, how scales are not provided, its not clear meaning of different colors.

The results are not well discussed with references to the previous studies.

Reviewer 3 Report

1). Figure 6 compares the EXP results and the FEM predictions regarding the contact-force for different impact energies. However, the curve deviation (EXP and FEM) for the 3 different cases is a little different. For 30J and 70J, the peak force matches great, but not for the 50J case. Can the author explain it?

2). In Figure 8, the author illustrates that in the EXP, the delaminate can only occur at the impact zone of the beam, while for the simulation, the delaminate occurs at both the impact zone and the corner. Can the authors explain this more? Is it due to the boundary condition setup?

3). Some typos being found:

-- Page 1, line 41, “most popularly” should be “most popular”

-- Page 2, line 50, “in the last years” should be “in the past years”

-- Page 11, line 336, “Figure 11” is duplicated; line 351, “Figure 12” is duplicated in the article.

Round 2

Reviewer 2 Report

Manuscripts can now be accepted in modified form.